# The Association between TNF-α, IL-6, and Vitamin D Levels and COVID-19 Severity and Mortality: A Systematic Review and Meta-Analysis

**DOI:** 10.3390/pathogens11020195

**Published:** 2022-02-01

**Authors:** Ceria Halim, Audrey Fabianisa Mirza, Mutiara Indah Sari

**Affiliations:** Faculty of Medicine, Universitas Sumatera Utara, Medan 20155, Sumatera Utara, Indonesia; ceriahalim@gmail.com (C.H.); audreyyfm@gmail.com (A.F.M.)

**Keywords:** TNF-α, IL-6, vitamin D, COVID-19

## Abstract

Background: An increasing number of scientific journals have proposed a connection between tumor necrosis factor-α (TNF-α) and interleukin-6 (IL-6) and the severity of COVID-19. Vitamin D has been discussed as a potential therapy for COVID-19 due to its immunomodulatory effects. This meta-analysis aims to determine the relationship, if any, between TNF-α, IL-6, vitamin D, and COVID-19 severity and mortality. Methods: The design of the study is a systematic review and meta-analysis. A literature search is performed using PubMed, Cochrane, ProQuest, and Google Scholar. Results: TNF-α insignificantly increases the risk of COVID-19 severity (adjusted odds ratio (aOR) = 1.0304; 95% CI 0.8178–1.2983; *p* = 0.80) but significantly increases the risk of COVID-19 mortality (crude hazard ratio (HR) = 1.0640; 95% CI 1.0259–1.1036; *p* = 0.0009). IL-6 significantly increases the risk of COVID-19 severity (aOR = 1.0284; 95% CI 1.0130–1.0441; *p* = 0.0003) and mortality (aOR = 1.0076; 95% CI 1.0004–1.0148; *p* = 0.04; adjusted hazard ratio (aHR) = 1.0036; 95% CI 1.0010–1.0061; *p* = 0.006). There is a statistically insignificant difference of the mean vitamin D levels between patients with severe COVID-19 and non-severe COVID-19 (mean difference (MD) = −5.0232; 95% CI 11.6832–1.6368; *p* = 0.14). A vitamin D deficiency insignificantly increases the risk of mortality of COVID-19 patients (aOR = 1.3827; 95% CI 0.7103–2.6916; *p* = 0.34). Conclusion: IL-6 is an independent prognostic factor towards COVID-19 severity and mortality.

## 1. Introduction

Coronavirus disease 2019 (COVID-19) was first reported in Wuhan, China. On 11 March 2020, the World Health Organization (WHO) declared COVID-19 to be a pandemic [1]. The three main symptoms of COVID-19 are a fever, cough, and shortness of breath. COVID-19 symptoms range from an asymptomatic infection to mild, moderate, severe, and critical such as acute respiratory distress syndrome (ARDS) [2]. Mortality has reached 39% to 45% in patients with ARDS [3,4]. In the United States, COVID-19 was the third most common cause of death in 2020, exceeding strokes and diabetes [5]. In Indonesia, the COVID-19 death toll reached 47,823 cases on 15 May 2021 [6].

Previous studies have proposed an association between immunopathogenesis and the clinical manifestation of COVID-19 [7,8,9]. A cytokine storm in COVID-19 patients causes apoptosis of the endothelial cells and epithelial cells as well as plasma leakage that can lead to lethal conditions such as ARDS, severe pneumonia, multiple organ failure, and shock [9,10,11].

Previous research has reported a significant difference of interleukin-6 (IL-6) levels between severe COVID-19 patients and non-severe COVID-19 patients [12,13,14,15,16,17,18]. Similar results were also observed on tumor necrosis factor-α (TNF-α) levels in COVID-19 patients [19,20,21,22,23].

Vitamin D has emerged as one of the proposed therapies for COVID-19. An association between COVID-19 severity and vitamin D deficiency (VDD) was reported by previous studies [24,25]. This finding was supported by previous studies that reported that vitamin D decreased the TNF-α and IL-6 concentration in the blood [26,27,28]. Other studies have reported the effect of vitamin D in increasing the Treg count and reducing inflammation [29,30].

This study aims to investigate whether TNF-α, IL-6, and vitamin D levels are associated with COVID-19 severity and mortality. This study differs from previous meta-analyses for the reason that this meta-analysis analyzes the odds ratios from logistic regression analyses and hazard ratios from Cox regression analyses. Furthermore, the odds ratios and hazard ratios are adjusted for other potentially confounding variables. In addition, this meta-analysis also includes the most recent studies up to the year 2021 to provide more comprehensive results.

## 2. Results

### 2.1. Literature Search

A total of 8802 records were identified across 4 databases using the search strategy described below. No additional articles or abstracts were selected from other sources. After 2202 duplicates were removed, 6600 citations were screened based on the title and abstract. Deduplication was carried out using the Rayyan application [31]. Five hundred and thirty-seven articles passed the initial screening and were further reviewed. Fourteen records did not have their full text available for retrieval. Five hundred and twenty-three full-text articles were reviewed and excluded based on the exclusion criteria. The exclusion reasons are shown in Figure 1. A total of 48 articles were included in the systematic review and meta-analysis with a sample size of 14,412 patients. 

### 2.2. Study Characteristics

Forty-eight studies were included in this meta-analysis. This meta-analysis analyzed the association between the levels of TNF-α, IL-6, and vitamin D and the severity and mortality of COVID-19. There were nine studies that analyzed the association between TNF-α and the severity and mortality of COVID-19 [32,33,34,35,36,37,38,39,40]. Along with those 9 studies, an additional 28 studies analyzed the association between the IL-6 level and the severity and mortality of COVID-19 [14,18,41,42,43,44,45,46,47,48,49,50,51,52,53,54,55,56,57,58,59,60,61,62,63,64,65,66]. A total of 11 studies analyzed the association between the vitamin D level and the severity and mortality of COVID-19 [67,68,69,70,71,72,73,74,75,76,77]. A Venn diagram is provided (Figure 2). The number of studies included in each analysis is listed in Figure 3. The number of patients in each analysis is listed in Figure 4.

The general characteristics of the studies are listed in Appendix A. All studies analyzed in this meta-analysis were published in 2020–2021. The dominant study location was China with a total of 27 studies (Figure 5a). Of the 48 included studies, the majority (26 studies) had a retrospective cohort study design (Figure 5b). A total of 22 studies analyzed the association between TNF-α, IL-6, and vitamin D levels and the severity of COVID-19. The most widely used severity criterion was the COVID-19 severity criterion from the Chinese National Health Commission, which was used by 11 studies (Figure 5c).

### 2.3. TNF-α and COVID-19 Severity

There were 6 studies that analyzed the association between the TNF-α level and severe COVID-19; these reported an odds ratio (OR) value from a logistic regression analysis with the TNF-α level as a continuous variable. The total sample was 1535 patients. The results of each study are listed in Appendix A.

A total of 3 studies reported results in the form of a crude OR. The results of the meta-analysis were that each increase in the TNF-α level of 1 pg/mL significantly increased the risk of developing severe COVID-19 (crude OR = 1.1004; 95% CI 1.0185–1.1890; *p* = 0.02). The forest plot can be in Figure 6. Four studies reported results in the form of an adjusted odds ratio (aOR). The meta-analysis showed that each increase in the TNF-α level of 1 pg/mL insignificantly increased the risk of developing severe COVID-19 (aOR = 1.0304; 95% CI 0.8178–1.2983; *p* = 0.80). The forest plot can be seen in Figure 7.

### 2.4. TNF-α and COVID-19 Mortality

Three studies analyzed the association between the TNF-α level and COVID-19 mortality; these reported an HR value from a Cox regression analysis with the TNF-α level as a continuous variable. The total sample was 1630 patients. The results of each study are listed in Appendix A. 

All three studies reported results in the form of a crude hazard ratio (HR). The results of the meta-analysis were that each increase in the TNF-α level of 1 pg/mL significantly increased the risk of mortality of COVID-19 patients (crude HR = 1.0640; 95% CI 1.0259–1.1036; *p* = 0.0009). The forest plot can be seen in Figure 8.

### 2.5. IL-6 and COVID-19 Severity

Sixteen studies analyzed the association between the IL-6 level and severe COVID-19; these reported OR values from a logistic regression analysis. The total sample was 6063 patients. The results of each study are listed in Appendix A.

A total of eight studies reported results in the form of a crude OR. The results showed that each increase in the IL-6 level of 1 pg/mL significantly increased the risk of developing severe COVID-19 (crude OR = 1.0643; 95% CI 1.0318–1.0979; *p* < 0.0001). The forest plot can be seen in Figure 9. Twelve studies reported results in the form of an aOR. The results of the meta-analysis were that each increase in the IL-6 level of 1 pg/mL significantly increased the risk of developing severe COVID-19 (aOR = 1.0284; 95% CI 1.0130–1.0441; *p* = 0.0003). The forest plot can be seen in Figure 10.

### 2.6. IL-6 and COVID-19 Mortality

Eleven studies analyzed the relationship between IL-6 and COVID-19 mortality; these reported OR values from a logistic regression analysis. The total sample was 2213 patients. The reported results of each study are listed in Appendix A.

Seven studies reported results in the form of a crude OR. The meta-analysis result was that each increase in the IL-6 level of 1 pg/mL significantly increased the risk of mortality of COVID-19 patients (crude OR = 1.0152; 95% CI 1.0067–1.0237; *p* = 0.0004). The forest plot can be seen in Figure 11. Eight studies published results in the form of an aOR. The result of the meta-analysis showed that each increase in the IL-6 level of 1 pg/mL significantly increased the risk of mortality of COVID-19 patients (aOR = 1.0076; 95% CI 1.0004–1.0148; *p* = 0.04). The forest plot can be seen in Figure 12.

Eleven studies analyzed the association between the IL-6 level and COVID-19 mortality; these reported HR values from a Cox regression analysis. The total sample was 6715 patients. The results of each study are listed in Appendix A.

A total of nine studies reported results in the form of a crude HR. The meta-analysis showed that each increase in the IL-6 level of 1 pg/mL significantly increased the risk of mortality of COVID-19 patients (crude HR = 1.0027; 95% CI 1.0013–1.0041; *p* = 0.0002). The forest plot can be seen in Figure 13. A total of seven studies reported results in the form of an adjusted hazard ratio (aHR). The result of the meta-analysis showed that each increase in the IL-6 level of 1 pg/mL significantly increased the risk of mortality of COVID-19 patients (aHR = 1.0036; 95% CI 1.0010–1.0061; *p* = 0.006). The forest plot can be seen in Figure 14.

### 2.7. Vitamin D and COVID-19 Severity

There were six studies that analyzed the association between the vitamin D level and severe COVID-19; these reported the mean vitamin D levels. The studies were assessed for quality using the Newcastle–Ottawa Scale. The total sample was 1424 patients. The results of each study are listed in Appendix A.

The result of the meta-analysis was that the vitamin D levels of severe COVID-19 patients were not significantly lower than those in non-severe COVID-19 patients (mean difference (MD) = −5.0232; 95% CI −11.6832–1.6368; *p* = 0.14). The forest plot can be seen in Figure 15.

### 2.8. Vitamin D and COVID-19 Mortality

Five studies analyzed the association between VDD and COVID-19 mortality; these reported OR values from a logistic regression analysis. VDD was defined as 25(OH)D < 20 ng/mL [78]. The total sample was 1339 patients. The results of each study are listed in Appendix A.

Three studies reported results in the form of a crude OR. The results of the meta-analysis showed that VDD insignificantly increased the risk of mortality of COVID-19 patients (crude OR = 1.1505; 95% CI 0.5299–2.4977; *p* = 0.72). The forest plot can be seen in Figure 16. All five studies reported results in the form of an aOR. The results were that VDD insignificantly increased the risk of mortality of COVID-19 patients (aOR = 1.3827; 95% CI 0.7103–2.6916; *p* = 0.34). The forest plot can be seen in Figure 17.

## 3. Discussion

TNF-α triggers and amplifies acute inflammatory reactions with characteristics of rubor, calor, dolor, and tumor [79]. TNF-α is one of the endogenous pyrogens that increases the body temperature and causes fever [80] (pp. 88–89). Large quantities of TNF-α are pathological to the organ systems. TNF-α can inhibit myocardial contractility and lower blood pressure, resulting in shock. TNF-α activates the tissue factor, which plays a role in the blood coagulation cascade [80] (p. 89).

The findings of this study were supported by the results of a previous study by Merza et al. [81], which reported that the mean TNF-α levels were not significantly higher in patients with severe COVID-19 compared with non-severe COVID-19 patients. A previous meta-analysis by Mulchandani et al. [82], however, reported that the mean TNF-α levels were significantly higher in patients with severe COVID-19 compared with non-severe COVID-19 patients. Several prior studies also presented that the TNF-α levels in severe COVID-19 patients were significantly higher than non-severe COVID-19 patients [19,21,83].

The mean TNF-α levels were found to be significantly higher in COVID-19 patients who died compared with COVID-19 survivors [84,85,86]. A previous study by Liu et al. [56] published that increased levels of TNF-α did not significantly reduce the risk of mortality in COVID-19 patients (aOR = 0.957; 95% CI 0.857–1.068; *p* = 0.429).

IL-6 is a pro-inflammatory cytokine that plays a role in the acute inflammatory reaction. IL-6 increases the leukocyte migration to areas of inflammation, increases the division of B cells that produce antibodies, and plays a role in Th17 differentiation in T cells [80] (pp. 85–87, 92). IL-6 activates the intracellular cascade of Jak/STAT (Janus kinases/signal transducers and activators of transcription) [87]. The inflammatory cascade of IL-6 through STAT3 forms a positive feedback loop—namely, the IL-6 amplifier (IL-6 AMP)—in non-immune cells. As a result, NF-κB hyperactivation occurs, which in turn produces an excessive number of cytokines, also known as a cytokine storm. A cytokine storm in COVID-19 patients is known to cause fatal conditions such as ARDS, severe pneumonia, multiple organ failure, and coagulation [10].

The mean IL-6 levels were recorded to be significantly higher in patients with severe COVID-19 compared with non-severe COVID-19 patients [15,16,88,89,90,91,92,93,94]. Additionally, IL-6 was also reported as an independent predictor of ICU admission [95]. The results of this meta-analysis were also in line with a previous meta-analysis of RCTs that presented a reduction in the incidence of intubation in COVID-19 patients receiving IL-6 inhibitors [96].

Higher mean IL-6 levels in COVID-19 deaths than in COVID-19 survivors were also reported by a previous meta-analysis [88]. Another study recorded that the mean IL-6 levels were 2.9 times higher in complicated COVID-19 patients compared with uncomplicated COVID-19 patients [97]. A meta-analysis of RCTs reported a decreased incidence of death in COVID-19 patients receiving IL-6 inhibitors [96].

Vitamin D is known to possess an anti-inflammatory effect. Vitamin D is reported to suppress the production of reactive oxygen species and myeloperoxidase in neutrophils as well as suppressing Th1 and Th17 activities, reducing the levels of pro-inflammatory cytokines, and increasing Th2 and Treg activities [26,29,30,98]. Vitamin D also inhibits the transcription factor NF-κB, which plays a role in the production of pro-inflammatory cytokines [98].

A previous study by AlSafar et al. [67] found VDD did not significantly increase the risk of severe COVID-19. Significantly lower vitamin D levels in severe COVID-19 patients than non-severe COVID-19 patients were reported by a prior meta-analysis [99]. VDD was associated with an increased risk of severe COVID-19 and vitamin D possessed a protective effect against the incidence of severe COVID-19 (OR = 0.91; 95% CI 0.84–0.99) [100]. VDD was estimated to cause a six-fold increased risk of developing severe COVID-19 [25].

No significant relationship was observed between an increased risk of mortality and VDD in studies using logistic regression analyses [100]. Another study noted no significant difference in the mean vitamin D levels between COVID-19 patients who died and COVID-19 survivors [101]. These results were in contrast to several other previous studies. VDD was estimated to cause a 15-fold increased risk of COVID-19 mortality [25]. A significant difference was recorded in the mean vitamin D levels between COVID-19 patients who died and COVID-19 survivors [73,77,102,103,104].

To our knowledge, the present meta-analysis—which included 14,412 patients from 48 studies—was the first study to analyze the OR and HR values between TNF-α, IL-6, and vitamin D and COVID-19 severity and mortality. The OR and HR values were more applicable to a clinical situation in calculating the risk of a patient rather than the mean difference values. Furthermore, these OR and HR values were adjusted for other variables whereas other studies using the mean did not. This meta-analysis also included studies up to the year 2021 to produce up-to-date results.

This study had several limitations. First, the differences between our results and the results of previous studies may be due to too few studies being included in our analysis. There are limited studies regarding COVID-19 due to the novelty of this disease. Our inclusion and exclusion criteria were more selective, resulting in fewer studies remaining after the selection process. For example, we excluded composite outcomes of severity and mortality to provide a better distinction between these two. There were also studies excluded due to different VDD cut-off values. These criteria ensured a more consistent result among the studies. Another limitation of this study was that the analyzed studies were mostly conducted in China. Thus, a potential bias due to a similar location and population cannot be ruled out.

## 4. Materials and Methods

A literature search was conducted on PubMed, Cochrane, ProQuest, and Google Scholar on 6 August 2021. The literature search was conducted with the keywords “COVID-19 AND (Tumor Necrosis Factor-alpha OR Interleukin-6 OR Vitamin D)”. The medical subject headings (MeSH) words and detailed search strategies are provided in Appendix A. The search results were compiled and deduplicated using Rayyan [31].

The studies included in this meta-analysis were studies with a population of COVID-19 patients and outcomes of severe COVID-19 and COVID-19 deaths. The inclusion criteria were: studies with a population of COVID-19 patients; available data of TNF-α, IL-6, and vitamin D levels in the blood; complete data were available; and the study was available in English. The exclusion criteria were: review articles; systematic reviews; meta-analyses; correspondence; letters; replies; comments; case reports; case series; paid journals; duplication; and research conducted on animals. In addition, studies that observed a composite severity outcome with mortality as one of the elements were excluded and regarded as studies with the wrong outcome. VDD was defined as 25(OH)D < 20 ng/mL [78]. Studies with different VDD cut-off points were excluded. The records were screened manually by one reviewer (CH).

All relevant data were collected using data collection standards set by two reviewers working together (CH and MIS). The collected data were the name of the first author, year of publication, number of patients, inclusion criteria, exclusion criteria, and levels of TNF-α, IL-6, and vitamin D. The data collected for TNF-α and IL-6 analyses were the OR from a logistic regression analysis or the HR from a Cox regression analysis. The data collected for vitamin D analyses were the mean and standard deviation (SD) or the OR from a logistic regression analysis. Missing and incomplete data were not sought further.

A quality assessment of the studies was carried out by two reviewers (CH and MIS). Studies with mean levels were assessed for quality using the NOS (The Newcastle–Ottawa Scale) [105]. The NOS criteria in the cross-sectional studies were adapted from the cohort criteria [106].

The odds ratio (OR), hazard ratio (HR), mean difference (MD), and 95% confidence interval (CI) were analyzed using Review Manager 5.4 (The Cochrane Collaboration, Oxford, UK). A *p*-value < 0.05 indicated statistically significant data. The heterogeneity of the statistical data was indicated by the symbol I^2^. An I^2^ value < 25% indicated a low heterogeneity, an I^2^ value of 25–50% indicated a moderate heterogeneity, and an I^2^ value > 50% indicated a high heterogeneity [107]. The data were analyzed using a random effects model. The random effects model was chosen because the studies in this meta-analysis were conducted by different researchers operating independently [108] (pp. 83–86). The random effects model was selected without considering the heterogeneity [109]. The study hypothesis was measured by the Z-test. Z-scores greater or less than ± 1.96 indicated that the results were significant [108] (pp. 257–258). All the results from the meta-analysis have been published in this article regardless of insignificant results.

## 5. Conclusions

The results of a literature search were 48 studies with a total sample of 14,412 patients. The majority of the included studies were located in China and the majority of the included studies had a retrospective cohort study design. Overall, the summary estimates of all analyses suggested that IL-6 was an independent prognostic factor toward COVID-19 severity and mortality. No definitive results were drawn regarding the association between TNF-α and vitamin D and COVID-19 severity and mortality. Further studies with more detailed information on other outcomes and cut-off values should be conducted especially in countries other than China to validate the association between these variables.

## Figures and Tables

**Figure 1 pathogens-11-00195-f001:**
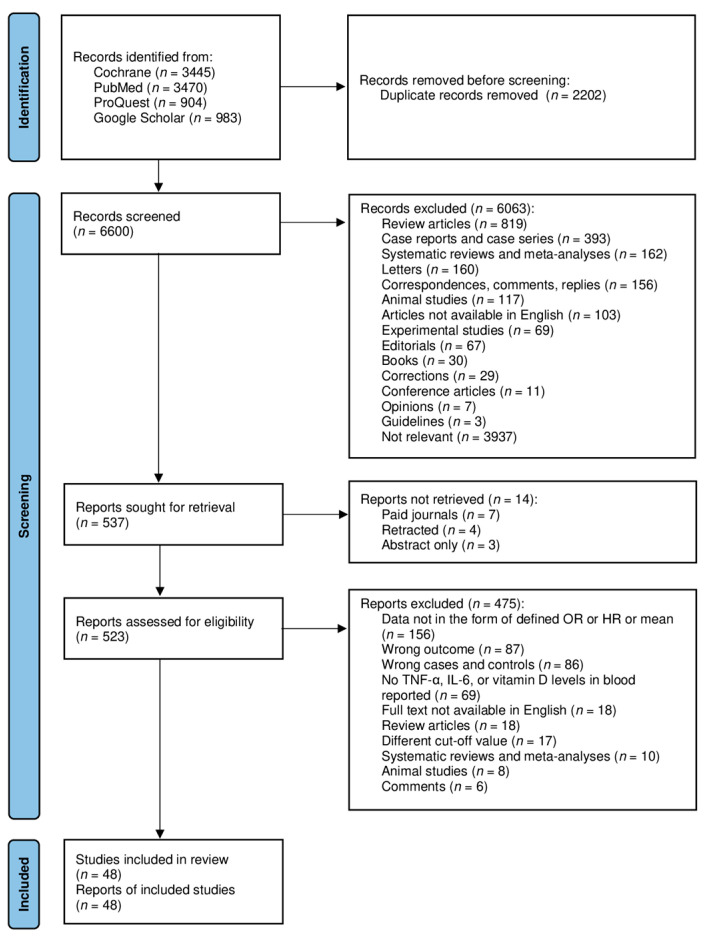
Article selection process flow diagram.

**Figure 2 pathogens-11-00195-f002:**
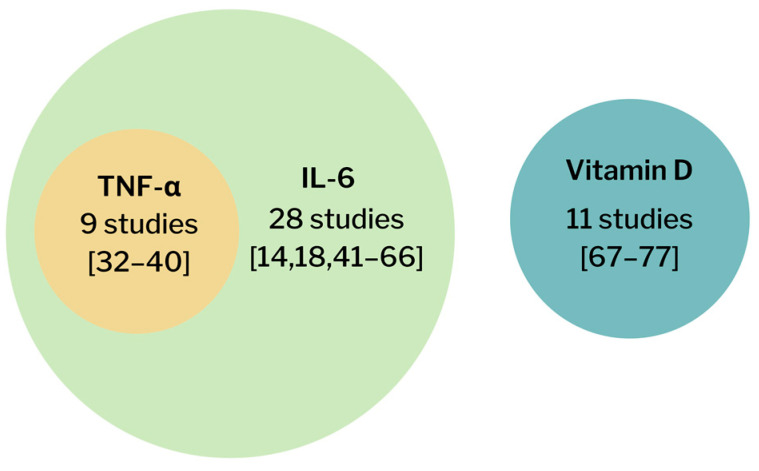
Venn diagram of the number of included studies.

**Figure 3 pathogens-11-00195-f003:**
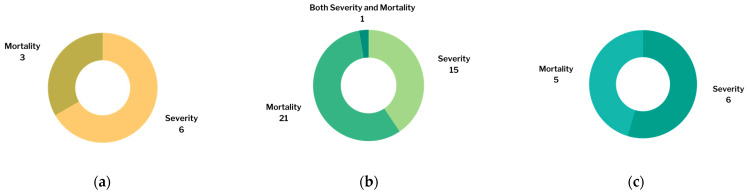
Number of studies included in: (**a**) TNF-α analysis; (**b**) IL-6 analysis; (**c**) vitamin D analysis.

**Figure 4 pathogens-11-00195-f004:**
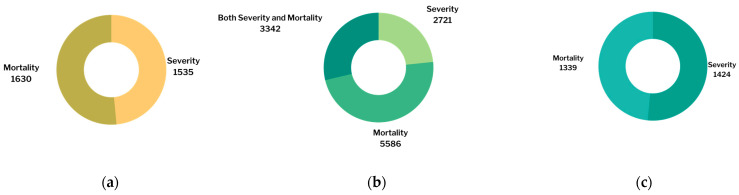
Number of patients included in: (**a**) TNF-α analysis; (**b**) IL-6 analysis; (**c**) vitamin D analysis.

**Figure 5 pathogens-11-00195-f005:**
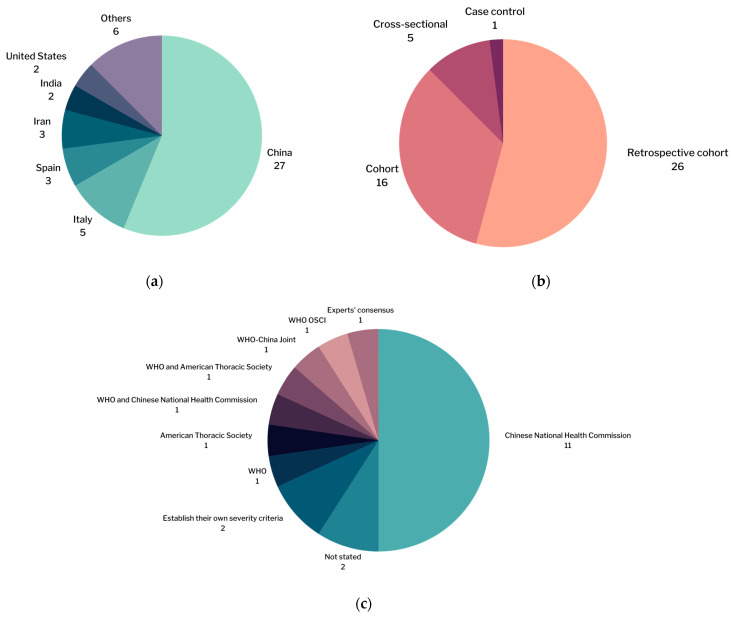
Number of studies by: (**a**) study locations; (**b**) study designs; (**c**) severity criteria used.

**Figure 6 pathogens-11-00195-f006:**
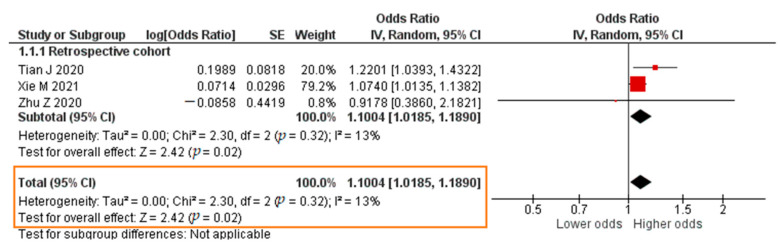
Forest plot of the crude odds ratio of severe COVID-19 with increased TNF-α levels. SE, standard error; IV, inverse variance; CI, confidence interval.

**Figure 7 pathogens-11-00195-f007:**
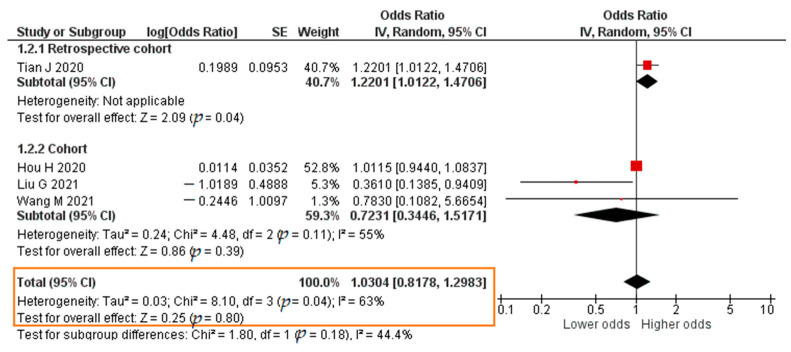
Forest plot of the adjusted odds ratio of severe COVID-19 with increased TNF-α levels. SE, standard error; IV, inverse variance; CI, confidence interval.

**Figure 8 pathogens-11-00195-f008:**
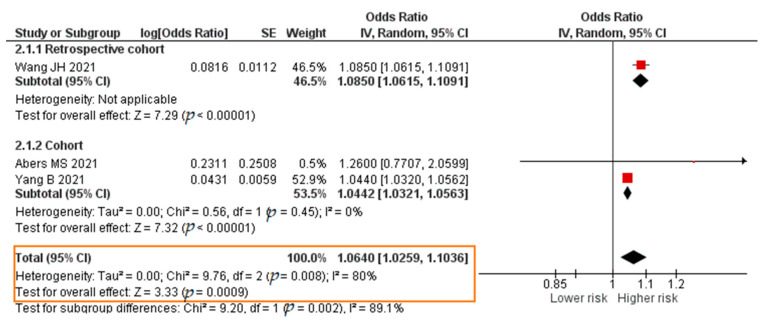
Forest plot of the crude hazard ratio of COVID-19 mortality with increased TNF-α levels. SE, standard error; IV, inverse variance; CI, confidence interval.

**Figure 9 pathogens-11-00195-f009:**
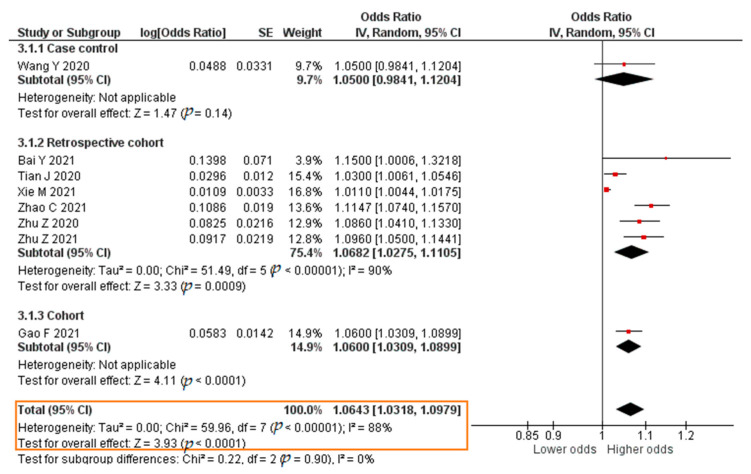
Forest plot of the crude odds ratio of severe COVID-19 with increased IL-6 levels. SE, standard error; IV, inverse variance; CI, confidence interval.

**Figure 10 pathogens-11-00195-f010:**
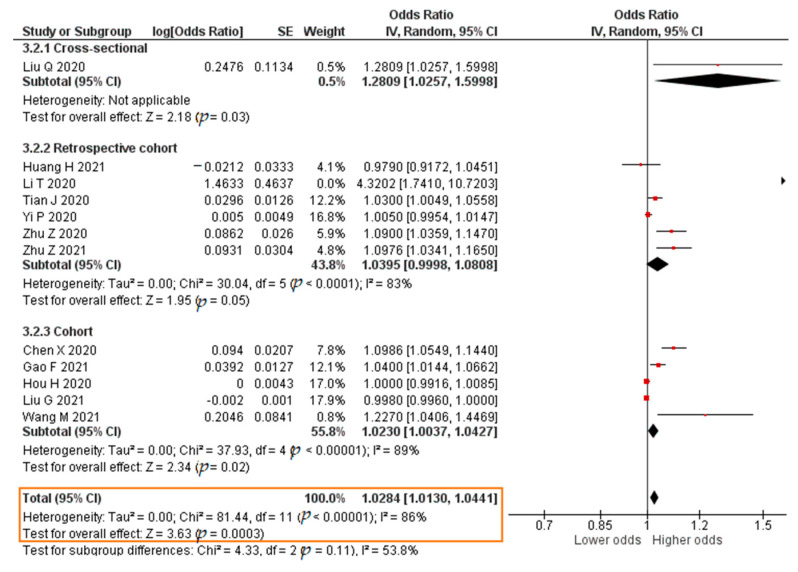
Forest plot of the adjusted odds ratio of severe COVID-19 with increased IL-6 levels. SE, standard error; IV, inverse variance; CI, confidence interval.

**Figure 11 pathogens-11-00195-f011:**
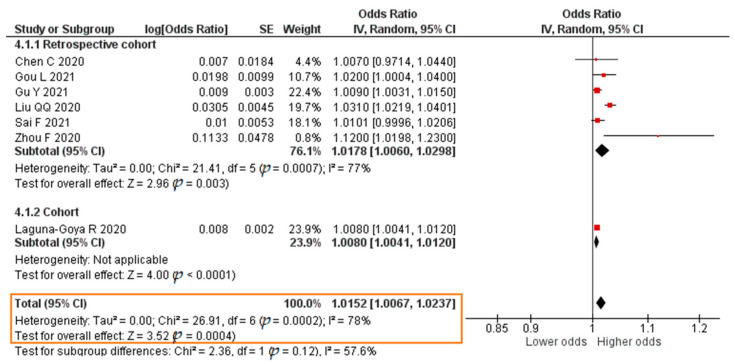
Forest plot of the crude odds ratio of COVID-19 mortality with increased IL-6 levels. SE, standard error; IV, inverse variance; CI, confidence interval.

**Figure 12 pathogens-11-00195-f012:**
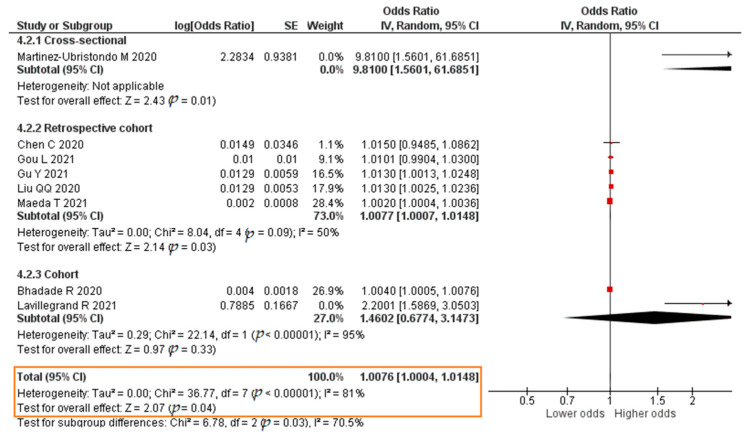
Forest plot of the adjusted odds ratio of COVID-19 mortality with increased IL-6 levels. SE, standard error; IV, inverse variance; CI, confidence interval.

**Figure 13 pathogens-11-00195-f013:**
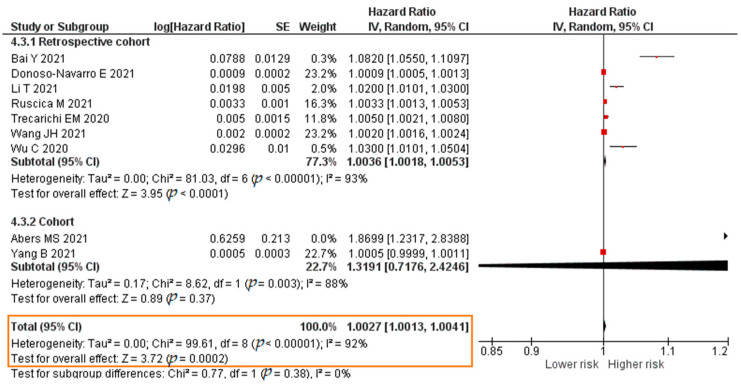
Forest plot of the crude hazard ratio of COVID-19 mortality with increased IL-6 levels. SE, standard error; IV, inverse variance; CI, confidence interval.

**Figure 14 pathogens-11-00195-f014:**
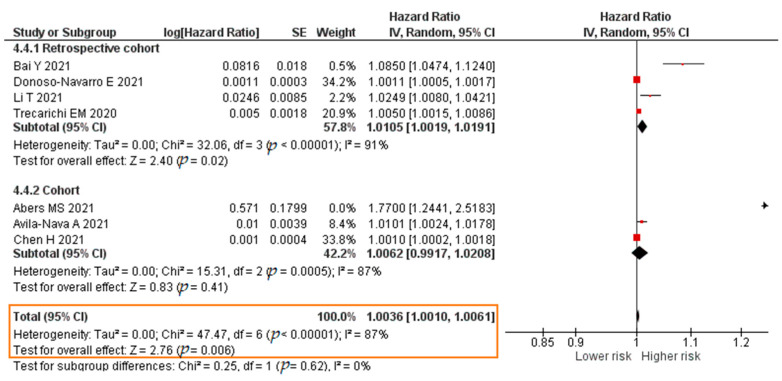
Forest plot of the adjusted hazard ratio of COVID-19 mortality with increased IL-6 levels. SE, standard error; IV, inverse variance; CI, confidence interval.

**Figure 15 pathogens-11-00195-f015:**
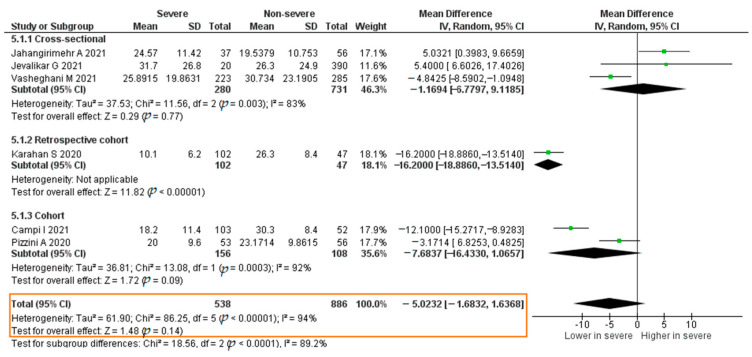
Forest plot of the mean difference of vitamin D levels between severe COVID-19 patients and non-severe COVID-19 patients. SE, standard error; IV, inverse variance; CI, confidence interval.

**Figure 16 pathogens-11-00195-f016:**
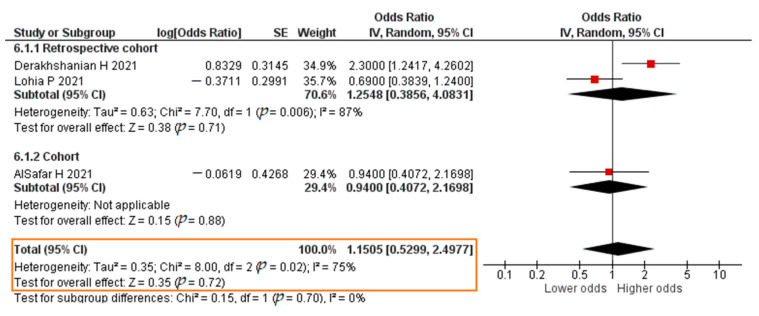
Forest plot of the crude odds ratio of COVID-19 mortality with vitamin D deficiency. SE, standard error; IV, inverse variance; CI, confidence interval.

**Figure 17 pathogens-11-00195-f017:**
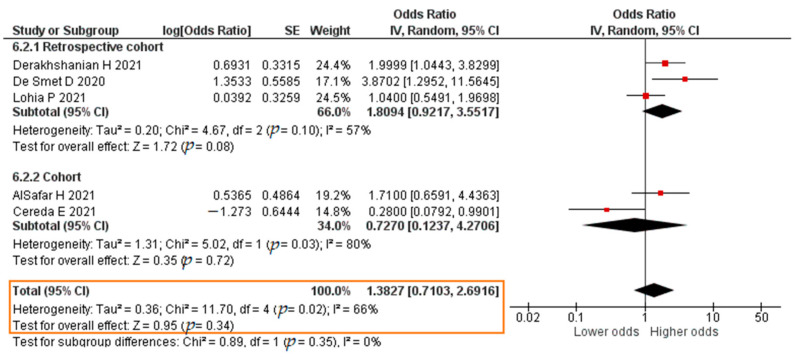
Forest plot of the adjusted odds ratio of COVID-19 mortality with vitamin D deficiency. SE, standard error; IV, inverse variance; CI, confidence interval.

## Data Availability

Data are contained within the article or Appendix A.

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
