# Peer review of "The Association between TNF-α, IL-6, and Vitamin D Levels and COVID-19 Severity and Mortality: A Systematic Review and Meta-Analysis"

_pathogens, 2022, doi:10.3390/pathogens11020195_

Round 1

Reviewer 1 Report

The authors of the manuscript: Association between TNF-α, IL-6, and Vitamin D Levels and 2 CoViD-19 Severity and Mortality: A Systematic Review and Meta-Analysis, performed a meta-analysis of several studies mostly originated from China.

Despite the fact that there are several studies in the litterature, focusing on this topic, the conclusions of this study are towards the correct direction. 

Major requests. 

  1. Please spent more ''manuscript'' space for the actual analysis that you are making, the forest plots etc and try to describe better the analysis and your findings, instead of giving all the ''manuscript ''space to the tables. Make the m/s more appealing to read. ie highlight your graphs, and put the actual statistics after the graph. 
  2. Write self explanatory legends and do not repeat the same words in subsequent paragraphs.
  3. Try to provide a good explanation on why your study is better than  previous studies, and the exclusion strategy you followed.

Minor requests. 

  1. Line 69 Results: Rephrase to make it clear to the reader to understand.
  2. Line 82: Are the included studies 44 or 48?
  3. Study characteristics: I would suggest to make a pie chart or something else to make it visual to the reader, to understand how many studies were included per topic and which was the total number of patients per study.
  4. Line 90:typo
  5. Discussion : Line 251:Please try to rephrase to make it more easy to understand: What does it mean inadequate studies?
  6. Lines 273-275, the IL-6 trypsin phrase is not relevant, please consider removing from the m/s.

Reviewer 2 Report

The study entitled "Association between TNF-α, IL-6, and Vitamin D Levels and CoViD-19 Severity and Mortality: A Systematic Review and Meta-Analysis" is an interesting topic. Background: More and more scientific journals have proposed the connection between tumor necrosis factor-α (TNF-α) and interleukin-6 (IL-6) to the severity of CoViD-19. Vitamin D has 
been discussed as a potential therapy for CoViD-19 due to its immunomodulatory effects. This meta-analysis aims to determine the relationship, if any, between TNF-α, IL-6, vitamin D and CoViD-19 severity and mortality. Methods: The design of the study is systematic review and meta-analysis. Literature search was done through Pubmed, Cochrane, Proquest, and Google Scholar. Results: TNF-α insignificantly increase risk of CoViD-19 severity (adjusted odds ratio (aOR)=1,0304; 95%CI 19
0,8178–1,2983; p=0,80), but significantly increase risk of CoViD-19 mortality (crude hazard ratio 20 (HR)=1,0640; 95%CI 1,0259–1,1036; p=0,0009). IL-6 significantly increase risk of CoViD-19 severity (aOR=1,0284; 95%CI 1,0130–1,0441; p=0,0003) and mortality (aOR=1,0076; 95%CI 1,0004–1,0148; 22
p=0,04; adjusted hazard ratio (aHR)=1,0036; 95%CI 1,0010–1,0061; p=0,006). There was a statistically insignificant difference of mean vitamin D levels between patients with severe CoViD-19 and non-severe CoViD-19 (mean difference (MD)=(−5,0232); 95%CI (−11,6832)–1,6368; p=0,14). Vitamin D deficiency insignificantly increase risk of mortality on CoViD-19 patients (aOR=1,3827; 95%CI 0,7103–2,6916; p=0,34). Conclusion: IL-6 is an independent prognostic factor towards CoViD-19 severity and mortality. However, the following should be addressed before submitting a revision

The authors stated, "Forty-eight studies were included in this meta-analysis" but Table 1 shows above 48 studies----need to check and include serial no in the table to avoid misinterpretation.

Which one is correct whether 9 or 6? include appropriate numbers
"There were 9 studies that analyzed the association between TNF-α and IL-6 levels and the severity and mortality of CoViD-19."

"There were 6 studies that analyzed the association between TNF-α level and the severity of CoViD-19 and reported odss ratio (OR) value from logistic regression analysis, with TNF-α level as a continuous variable" 

Is it 48 or 44 in total?
"Of the 44 included studies, the majority had a retrospective cohort study design, which consisted of 26 studies"

The discussion should be improved. 

The meta-analysis needs critical revision, especially on included studies that make a major difference in the entire results. 

Round 2

Reviewer 2 Report

Accept in present form